# Implantation and Decidualization in PCOS: Unraveling the Complexities of Pregnancy

**DOI:** 10.3390/ijms25021203

**Published:** 2024-01-18

**Authors:** Satoko Matsuyama, Sarah Whiteside, Shu-Yun Li

**Affiliations:** Reproductive Sciences Center, Division of Developmental Biology, Cincinnati Children’s Hospital Medical Center, Cincinnati, OH 45229, USA; satoko.matsuyama@cchmc.org (S.M.); sarah.whiteside@cchmc.org (S.W.)

**Keywords:** PCOS, implantation, decidualization, pregnancy, animal model

## Abstract

Polycystic Ovary Syndrome (PCOS) is a prevalent endocrine disorder in women of reproductive age, affecting 5–15% globally with a large proportion undiagnosed. This review explores the multifaceted nature of PCOS and its impact on pregnancy, including challenges in fertility due to hormonal imbalances and insulin resistance. Despite restoring ovulation pharmacologically, women with PCOS face lower pregnancy rates and higher risks of implantation failure and miscarriage. Our review focuses on the complexities of hormonal and metabolic imbalances that impair endometrial receptivity and decidualization in PCOS. Disrupted estrogen signaling, reduced integrity of endometrial epithelial tight junctions, and insulin resistance impair the window of endometrial receptivity. Furthermore, progesterone resistance adversely affects decidualization. Our review also examines the roles of various immune cells and inflammatory processes in the endometrium, contributing to the condition’s reproductive challenges. Lastly, we discuss the use of rodent models in understanding PCOS, particularly those induced by hormonal interventions, offering insights into the syndrome’s impact on pregnancy and potential treatments. This comprehensive review underscores the need for advanced understanding and treatment strategies to address the reproductive complications associated with PCOS, emphasizing its intricate interplay of hormonal, metabolic, and immune factors.

## 1. Introduction

Polycystic Ovary Syndrome (PCOS) is a highly prevalent endocrine disorder affecting women of reproductive age, characterized by a combination of clinical, hormonal, and morphological anomalies. Worldwide, PCOS affects approximately 5–15% of women, and up to 70% of cases are undiagnosed, making it a significant concern for both patients and healthcare professionals [1]. Its etiology is intricate, encompassing genetic and epigenetic susceptibility, intricacies within the hypothalamic pituitary axis and ovarian dysfunction, excess androgen exposure, insulin resistance, and adiposity-related mechanisms [2]. The Rotterdam Criteria define the diagnosis of PCOS through three main features: hyperandrogenism (either clinical signs like hirsutism or biochemical markers), ovulatory dysfunction, and polycystic ovaries visible on ultrasound [3].

PCOS women have an approximately 40% increase in gonadotropin hormone-releasing hormone (GnRH) pulse frequency [4]. This rapid secretion of GnRH stimulates the pituitary gland to overproduce LH, prompting increased production of androgens from the ovaries. Concurrently, follicle-stimulating hormone (FSH) concentrations often remain at average or decreased levels, hindering the proper maturation of eggs. As a result, this dysfunction in hormonal regulation culminates in the formation of multiple small cysts and irregularities in ovulation. The development of ovarian cysts then contributes to hormone imbalance and impaired ovulation, and leads to increased levels of AMH. Elevated AMH further boosts GnRH’s activity, which in turn amplifies the production of LH and thus the synthesis of androgens. Additionally, a significant proportion of women with PCOS have insulin resistance, manifesting in heightened insulin levels [2]. Hyperinsulinemia can stimulate theca cells, endocrine cells located in the ovarian follicles, to produce more androgens, creating a positive feedback loop in which hormonal and metabolic perpetuations continually exacerbate one another [5].

Delving deeper into the intricacies of PCOS reveals a multitude of complexities surrounding pregnancy. For many women with PCOS, fertility is a significant challenge, often necessitating a comprehensive understanding of their health history and tailored interventions. The syndrome significantly affects the quality and regularity of ovulation. Anovulation or irregular ovulation often becomes the norm rather than the exception for many women with this condition, posing additional challenges for those desiring pregnancy [1]. One of the primary reasons for fertility challenges is the hormonal imbalances inherent in PCOS. Elevated levels of androgens play a pivotal role in disrupting the regular ovulatory cycle, thereby influencing fertility [6]. Insulin resistance and associated metabolic abnormalities are not just mere bystanders in this condition. A significant proportion of women with PCOS exhibit insulin resistance, which exacerbates hormonal imbalances while directly affecting the quality of the ovum, thus adding to fertility challenges [7,8,9].

The primary treatment choice for infertility in patients with PCOS is lifestyle modification, which includes nutritional improvements, increased physical activity, and possibly weight loss [10]. The potential benefits of using metformin [11] or glucagon-like peptide 1 receptor agonists (GLP-1RAs) agonists [12] are also noted. Typically, the first-line pharmacological treatment for infertility in PCOS patients involves oral ovulation inducers such as clomiphene citrate and letrozole. Clomiphene citrate, a selective estrogen receptor modulator (SERM), functions by inhibiting the negative feedback of estrogen in the hypothalamus, leading to increased FSH secretion and consequent follicular growth and ovulation [10]. On the other hand, letrozole, an aromatase inhibitor (AI), works by inhibiting the conversion of androgens to estrogens, thereby decreasing negative feedback in the hypothalamus and increasing FSH secretion [10,13]. Letrozole should be used rather than clomiphene citrate in women with PCOS with anovulatory infertility and no other infertility factors to improve ovulation, clinical pregnancy, and live birth rates [14]. If patients do not respond to lifestyle modifications or oral ovulation induction with SERM or AI, exogenous gonadotropins may be considered as a treatment option [10]. In cases where oral treatments fail to induce ovulation, or several cycles of ovulation do not result in pregnancy, in vitro fertilization (IVF) should be considered. IVF has been shown to be effective in PCOS patients, often resulting in an increased number of retrieved mature oocytes and higher clinical pregnancy rates [10]. However, it is important to assess and manage the risk of ovarian hyperstimulation syndrome (OHSS) when planning IVF cycles in PCOS patients [15].

Even after ovulation is pharmacologically restored, many anovulatory patients still encounter significant hurdles. These patients experience reduced cumulative pregnancy rates and face a higher risk of implantation failure and spontaneous miscarriage. These complexities underscore the urgent need for a deeper understanding of the underlying mechanisms of PCOS. Developing more effective treatment strategies is paramount to improve reproductive outcomes and ensure the overall health of women battling this condition. As we venture further into this review, we will unravel the multifaceted nature of PCOS, lay the foundation of its effects on pregnancy, and explore the numerous challenges it presents.

## 2. Implantation in PCOS

### 2.1. Endometrial Receptivity and Implantation Challenges

PCOS is characterized by implantation challenges due to both hormonal and metabolic imbalances. Women with PCOS often experience abnormal increases in LH during the follicular phase, which, along with excess androgen, triggers apoptosis and autophagy in granulosa cells. This triggers oocyte arrest and anovulation, diminishing the quality of oocytes and embryos, and impairing ovarian function [16,17]. For women struggling with ovulation, in vitro fertilization/intra-cytoplasmic sperm injection (IVF/ICSI) stands as a tertiary treatment option. Despite selecting high-quality embryos for transfer, women with PCOS still face a range of negative pregnancy outcomes, including lower embryo implantation rates and increased occurrences of early clinical pregnancy loss [18,19].

The hormonal imbalances in PCOS, particularly hyperandrogenism and altered LH/FSH ratios, disrupt the endometrial cycle [20], leading to decreased levels of progesterone and leukemia inhibitory factor (LIF), both crucial for endometrial receptivity and implantation [21,22]. Hyperandrogenism in PCOS specifically inhibits the growth of endometrial cells and their decidualization, negatively impacting embryo implantation. This issue is compounded by the dysregulation between androgen receptors and Wilms tumor protein (WT1), which may further reduce endometrial receptivity [23]. Insulin resistance, another feature of PCOS, disrupts glucose metabolism within the endometrium, contributing to progesterone resistance and exacerbating issues with endometrial functionality. This heightens the risk of miscarriages and increases the likelihood of atypical hyperplasia and endometrial cancer. The complex manifestations of these dysfunctions underscore the necessity for a nuanced approach in treating the reproductive aspects of PCOS, one that addresses the complex interplay between hormonal, metabolic, and endometrial factors [24,25,26]. Thus, various signaling pathways such as hyperandrogenism, LIF, and insulin resistance are disrupted in PCOS, leading to impaired endometrial receptivity (Figure 1).

### 2.2. Disrupted Estrogen Signaling in PCOS

A receptive endometrium is prepared through the sequential exposure to estrogen and progesterone. Estrogen prompts the proliferation of the endometrial lining during the proliferative phase and enhances the expression of progesterone receptor (PGR). However, excessive estrogen activity can impair endometrial receptivity. Estrogen receptor alpha (ERα/ESR1) increases in response to estrogen during the proliferative phase of the menstrual cycle, but progesterone-induced down-regulation of ESR1 is crucial during the secretory phase for embryo implantation. After ovulation, progesterone triggers significant cellular changes to create a receptive endometrium and maintain early pregnancy [27]. Although estrogen levels are within the normal range in PCOS patients, there are decreased progesterone levels due to oligo-/anovulation which disrupts this estrogen–progesterone balance, impacting fertility and increasing miscarriage risk. Furthermore, research on ESR1 knockout and ESR2 knockout mouse models has confirmed the importance of ESR1 in endometrial receptivity [28]. In both PCOS patients and animal models, ESR1 is significantly increased in the endometrium, resulting in up-regulation of Ki-67 in both epithelium and stroma [18,29]. In women with unexplained infertility, the expression of ESR1 is also significantly elevated in the endometrium during the mid-secretory phase (implantation window) [30]. Furthermore, an increased proliferation index, due to higher estrogen levels in the endometrium, is associated with infertility [31].

LIF, secreted by the glandular epithelium in response to estrogen, plays a critical role in implantation. It triggers the activation of Janus kinase (JAK), a tyrosine kinase without a receptor, which in turn activates the signal transducer and activator of transcription 3 (STAT3) through phosphorylation [32]. Mice lacking LIF exhibit no expression of epidermal growth factor (EGF)-like growth factors such as heparin-binding epidermal growth factor (*Hbegf*), amphiregulin (*Areg*), and epiregulin (*Ereg*) around the blastocyst on day 4 of gestation [33,34]. Additionally, mice deficient in STAT3 show heightened epithelial expression of estrogen-responsive genes like lactotransferrin (*Ltf*) and Mucin1 (MUC1), leading to increased estrogen signaling and sustained proliferation in the luminal epithelium. This over-proliferation, coupled with growth inhibition of the stromal layer, points to a uterine state that is not conducive to implantation [35,36]. These results together suggest that disruption of the LIF-STAT3 pathway results in an undifferentiated uterine epithelium, rendering it inhospitable to embryonic implantation. In PCOS mouse models, the LIF-STAT3 pathway is inactivated in the uterus during the implantation window, resulting in an induction of *Ltf* and MUC1 [36]. MUC1, a transmembrane mucin found on the apical side of uterine epithelial cells, is a protective barrier against microbial invasion and enzyme degradation. The reduction of MUC1 is considered essential for establishing a functionally receptive uterine environment in various species [37]. In ovulated PCOS women, MUC1 is strongly expressed in the endometrial epithelium compared to healthy women, suggesting that endometrium is in a non-receptive status [38].

In the endometrium, Wnt/β-catenin signaling contributes to the regulation of cellular proliferation and differentiation throughout the menstrual cycle. During the cycle’s proliferative phase, estradiol activates Wnt/β-catenin signaling, whereas in the secretory phase, progesterone suppresses this pathway, thereby moderating the proliferative effects of estrogen [39]. In PCOS women, there is a notable increase in β-catenin in the endometrium during the proliferative phase [40,41]. β-catenin is involved in the regulation of numerous genes, including Msx1, which has been implicated in endometrial receptivity—a critical factor in successful embryo implantation [42]. The correlation between increased levels of β-catenin and Msx1 suggests a potential mechanism by which PCOS influences endometrial receptivity. In PCOS mouse models, the mRNA level of Msx1 has been shown to be significantly increased in both luminal and glandular epithelium during the implantation window, suggesting that the dysregulation of Wnt/β-catenin/Msx1 compromises endometrial receptivity in PCOS [36].

### 2.3. Indian Hedgehog Signaling in PCOS

Indian hedgehog (*Ihh*), a member of the hedgehog gene family, is produced in the epithelium under the regulation of progesterone and regulates stromal cell function through paracrine actions [43,44]. Research using a conditional *Ihh* knockout mouse model showed that without *Ihh*, the uterus becomes nonreceptive due to inhibited stromal cell growth and vascular development. This can be coupled with enhanced estrogen signaling during the peri-implantation period [45]. Studies have also revealed a marked elevation in the mRNA levels of *Ihh* within the luminal and glandular epithelia of the endometrium during the implantation window in a PCOS mouse model [36]. However, the mechanism underlying how this abnormal expression of *Ihh* impairs endometrial receptivity is not clearly defined.

### 2.4. Decreased Endometrial Epithelial Tight Junctions in PCOS

Elevated testosterone levels influence the integrity of tight junctions in the endometrial epithelium by decreasing Claudin 4 (Cldn4) and Occludin, both of which are integral proteins for maintaining tight junction integrity. This hormonal effect may have significant implications for the process of embryo attachment and subsequent implantation [46]. There is a notable reduction in Cldn4 and Cldn1 in the endometrial epithelium in PCOS, contributing to compromised intercellular junctions. This disruption might further impair the endometrial receptivity [40,47].

### 2.5. Insulin Resistance Impairs Endometrial Receptivity in PCOS

Hyperinsulinemia and insulin resistance are common features of PCOS. Excess insulin disrupts the hormonal equilibrium vital for the endometrium’s proper functioning. The corresponding insulin resistance in hyperinsulinemia further aggravates the situation by demanding higher insulin levels for maintaining glucose balance. Consequently, there is a disturbance in the expression of key molecules for endometrial receptivity. Consequently, hyperinsulinemia is associated with decreased activation of insulin receptor substrate 1 (InRS-1) and impaired glucose transport in the endometrial stroma, potentially through inflammatory pathways [19,48].

Mice on a high-fat diet (HFD) that developed insulin resistance showed that this condition impairs uterine receptivity [49]. There is a notable decrease in the mRNA levels of *Lif*, *Msx1*, and *Cldn4*, suggesting compromised endometrial receptivity. Mitochondrial dysfunction and endoplasmic reticulum stress are features of peripheral insulin resistance [49]. Mitochondria play a key role in energy production, and mitochondrial dysfunction at the cellular level can affect systemic metabolic balance. There was a notable decrease in mitochondrial DNA within the uterus and an interruption in the normal function of mitochondria in insulin resistance mice during the implantation window. However, the gene expression associated with endoplasmic reticulum stress remained unchanged. This suggests that insulin resistance-induced mitochondrial dysfunction uniquely influenced the oxidative stress response, impairing the endometrial receptivity in the uterus [49]. In a PCOS-like pregnant rat model, there was noticeable swelling of the mitochondria within the stromal tissue. These mitochondria displayed a loss of their typical tubular structure and the internal folds known as cristae were indistinct and collapsed within the decidual stromal cells, indicating a mitochondrial dysfunction in stromal cells during pregnancy [50]. The recent wide acceptance of functional mitochondrial disorders as a correlated factor of numerous diseases has led to the presupposition that abnormal mitochondrial metabolic markers are associated with PCOS [51].

## 3. Decidualization in PCOS

Decidualization is a critical step of the endometrial cycle that creates a receptive microenvironment and prepares the uterine lining for the establishment of the placenta [52]. The human endometrium undergoes a cyclical regeneration during the menstrual cycle, triggered by fluctuating levels of estrogen and progesterone. During the mid-secretory phase, when progesterone levels are high, human endometrial stromal cells (hESCs) undergo a transformation. They differentiate to large epithelioid-like decidual cells through a process known as the decidual reaction. These decidualized cells establish an environment suitable for successful implantation of the embryo. Notably, human decidualization is embryo-independent, initiates spontaneously, and occurs in the secretory phase of the menstrual cycle [52]. Conversely, decidualization occurs after embryo implantation in mice [53]. Decidualization is controlled by the complex interplay of hormones and signaling cascades. Any disruptions in these interactions can compromise the decidualization process, resulting in a myriad of pregnancy disorders including infertility, miscarriage, and uteroplacental disorders. Overall, decidualization plays a critical role in both humans and mice, contributing to proper embryo implantation, development, and a successful pregnancy [54]. 

Decidualization regulates placental formation by controlling the invasiveness of trophoblast cells from the blastocyst. The maternal decidua secretes cytokines and growth factors to promote trophoblastic invasion [55]. The trophoblasts then adhere and invade into the maternal endometrium and form both the placenta and extraembryonic membranes. Furthermore, through cultivating a rich vascular network, the decidua supports nutrients, gas, and waste exchange between the fetus and mother [55]. Placentation also requires active tolerance mechanisms to prevent the rejection of the conceptus. The maternal decidua contains maternal immune cells which act as a barrier and establish defense against potential pathogens. Natural killer cells and macrophages also surround the invading trophoblast cells, aiding in the establishment of the placenta [56]. In sum, decidualization provides a nurturing environment for the developing embryo, facilitates nutrient exchange, and prevents immune rejection of the embryo. Likewise, inadequate decidualization of the endometrium can lead to abnormal implantation/placentation and adverse pregnancy outcomes [54].

### 3.1. Decidualization Defect in PCOS

Hyperandrogenism serves as the distinct biochemical feature of PCOS patients. Elevated androgen inhibits the proliferation and differentiation of endometrial stromal cells, resulting in incomplete decidualization, and thereby disrupting embryo implantation [57]. Due to dysregulated hormonal and molecular signaling pathways, cells derived from PCOS biopsies exhibit a delayed and incomplete morphological change during in vitro decidualization. Hyperandrogenism also impairs the normal function of the corpus luteum, leading to reduced progesterone levels [20]. Furthermore, progesterone resistance, likely stemming from insulin resistance, contributes to reduced decidualization [20]. Many PCOS patients experience progesterone resistance, with their tissues becoming less responsive to progesterone due to long-term anovulation, contributing to impaired decidualization [20]. Progesterone resistance in PCOS also results in lower anti-androgen activity, inflammation of the endometrium, and compromised stromal differentiation, elevating the risk of disrupted decidualization [58].

Women with PCOS experience a myriad of pregnancy complications as a consequence of impaired decidualization which disrupts the maternal response to hormonal signaling. Recent studies indicate that women with compromised cyclic decidualization have an increased susceptibility to pregnancy failure [59]. PCOS patients often encounter challenges in conceiving and have pregnancy-related complications [60]. For instance, PCOS patients have an increased risk of miscarriages, uteroplacental disorders, and preterm birth. PCOS patients also display a three- to four-fold increased risk of developing pregnancy-induced hypertension or preeclampsia [61]. Overall, the pathogenesis and downstream consequences of PCOS are associated with impaired molecular signaling pathways and sex steroid hormone receptors, include estrogen receptors (ER), progesterone receptors (PR), and androgen receptors (AR) (Figure 2).

### 3.2. Altered Progesterone Signaling in PCOS

Progesterone is a steroid hormone that orchestrates the endometrium’s transition into the secretory phase and remains indispensable for endometrial decidualization [62]. In a typical menstrual cycle, progesterone levels rise and fall, marking the ebb and flow of endometrial differentiation and maturation. However, in PCOS, due to long-term anovulation, this rhythmic interplay is disrupted. The endometrium, deprived of the regulatory influence of progesterone, suffers from altered periodicity of progesterone levels [20]. Insufficient or absent ovulation leads to reduced progesterone levels. Luteal phase deficiency is a condition characterized by low progesterone levels and limited uterine lining growth, resulting in implantation failure. Progesterone supplementation is the most common treatment for luteal phase deficiency [63]. Cyclic progesterone therapy also improves androgenic PCOS, showing reduced GnRH pulsatility, lowering androgens, and promoting ovulation [64,65].

Progesterone signaling is crucial for decidualization; progesterone acts via its nuclear receptor (PGR) to transcriptionally reprogram the hESCs during the decidual reaction. P4 is secreted from the corpus luteum following ovulation and induces the proliferation and decidualization of ESCs. The PGR signaling network is composed of multiple signaling pathways and downstream regulators including heart and neural crest derivatives-expressed(HAND2), Forkhead Box O1 (FOXO1), bone morphogenesis protein 2 (BMP2), and wingless-related integration site (WNT) signaling, among others [54]. While progesterone levels are lower than normal in patients with PCOS, endometrial cells from PCOS patients display increased PGR expression [29]. Some studies suggest this phenomenon is explained by progesterone resistance as one study reported that PGR-mediated signaling pathways in the nucleus are associated with progesterone resistance in PCOS [66].

HOX gene expression is regulated by sex steroids; in both humans and mice, HOXA10 is primarily controlled by P4 in the endometrium. HOXA10 is a developmentally regulated homeobox transcription factor directly involved in decidualization and highly expressed in stromal cells during the decidual reaction. In humans, HOXA10 mRNA expression is increased in the decidualizing endometrial stroma during the secretory phase. However, the expression of HOXA10 is altered in the endometrium of PCOS patients [67]. In endometrial biopsies obtained from women with PCOS, elevated androgen secretion and aberrant progesterone signaling decreased HOXA10 expression during the secretory phase of the menstrual cycle compared with normal fertile women, leading to impaired decidualization [67]. Similarly, targeted deletion of HOXA10 in mice results in female infertility due to severe decidualization defects [67]. HOXA10 mutant mice have diminished stromal cell responsiveness to progesterone, impaired stromal cell proliferation/differentiation, and compromised implantation sites causing defective decidualization [68].

Promyelocytic leukemia zinc finger (PLZF), a progesterone-dependent transcription factor, plays an important role in decidualization. PLZF drives progesterone-dependent transcriptional reprogramming of the human endometrial stromal cells to promote decidualization. Specifically, PLZF regulates genes involved in cellular motility and proliferation required for the transformation of hESCs into epithelioid decidual cells [69]. Both activations of AR and PGF induce PLZF expression, while PLZF acts as a negative feedback regulator of AR. Furthermore, PLZF knockdown in mice impairs cell adhesion, ECM remodeling, cell migration, and hormonal responsiveness, all contributing to compromised decidualization [69].

HAND2 is another progesterone-dependent transcription factor that plays a role in the progesterone-induced decidualization of hESCs. Furthermore, HAND2 regulates downstream expression of forkhead box protein O1 (FOXO1) [52]. In healthy women, HAND2 expression significantly increases in the secretory phase during the decidualization of the stromal cells. Suppressing HAND2 expression in both mice and human endometrial cells decreases the expression of decidualization markers [70]. Furthermore, FOXO1 gene expression has been found to be downregulated in the ovaries of women with PCOS, contributing to the distinctive manifestations of PCOS [71].

### 3.3. cAMP Signaling in PCOS

The intracellular second messenger, cyclic adenosine monophosphate (cAMP), is stimulated by G protein-coupled receptors and regulates a multitude of cellular responses and intracellular events [72]. The PGR responds to progesterone and cyclic AMP/protein kinase A (PKA) during decidualization. cAMP is a known inducer of decidualization in human cell culture systems and increases the expression of decidual markers [52]. cAMP sensitizes ESCs in the endometrium to progesterone and stimulates the transcriptional activity of PGR. cAMP-induced signaling pathways are important for the process of endometrial decidualization as they activate downstream pathways and induce FOXO1 expression. Furthermore, recent studies show that women with PCOS have reduced cAMP-mediated PKA signaling in stromal cells, potentially contributing to impaired decidualization [73].

cAMP signaling also has a major impact on ESR1-dependent and PR-dependent expression of genes involved in decidualization [52]. ESR1 controls expression of Wnt Family Member 4 (WNT4), FOXO1, and PGR in the endometrium. During hESC differentiation, cAMP-dependent PKA phosphorylates Mediator 1 (MED1), a subunit of the mediator coactivator complex. MED1 then interacts with ESR1 to regulate downstream expression of WNT4, FOXO1, and PGR. In vitro studies reveal that expression of FOXO1 and WNT4 is down-regulated in ESR1-depleted hESC during the differentiation process, due to the loss of MED1 activity [74].

### 3.4. Aberrant Estrogen Signaling in PCOS Patients

PCOS is associated with abnormal functioning of the female sex hormone estrogen and estrogen receptors (ERs) [75]. The cyclic secretion of estrogen in the menstrual cycle modulates ovarian function while regulating the growth and proliferation of the endometrium. Estrogen acts via ESR1 and ESR2 pathways; the estrogen + ER complex translocates to the nucleus and alters the transcription of genes involved in decidualization. Estrogen signaling recruits second messengers, including cAMP, to induce decidualization. PGR expression is also induced by E2 action through ESR1, and PGR can then feedback and inhibit ESR1 expression during the luteal phase, creating a fine-tuned system to balance hormonal effects [76]. Likewise, disturbed hormonal levels in the menstrual cycle dysregulate decidualization and contribute to pregnancy-related complications. Progesterone is secreted by the corpus luteum during the luteal phase of the menstrual cycle and inhibits estrogen-induced proliferation to provide an appropriate uterine environment for the embryo. Likewise, the expression of estrogen receptors decreases in the secretory phase of healthy women [59]. Due to aberrant progesterone signaling in PCOS, this process is impaired and estrogen’s antagonistic ability is diminished due to low progesterone levels. ESR1 expression in the endometrium is enhanced in the secretory phase among patients with PCOS compared to healthy women [74].

### 3.5. The Role of Androgen Receptors in PCOS-Related Complications

The endometrium is an androgen target tissue that expresses androgen receptor (AR). Androgens are important players in endometrial physiology and contribute to the development of a suitable microenvironment for the conceptus [59]. PCOS is characterized by aberrant expression of the AR in the endometrium. In healthy females, AR expression is upregulated in the mid/late secretory phase and androgens stimulate the proliferation of endometrial stromal cells through AR activation [59]. However, hyperandrogenism in PCOS delays decidual transformation of endometrial cells, likely due to the overexpression of androgen receptors in women with PCOS. Moreover, MAGE Family Member A11 (MAGEA11), a co-regulator of AR, is constitutively overexpressed in PCOS patients. Enhanced MAGEA11 and AR-mediated transcriptional regulation impair the decidual reaction [57]. Furthermore, a number of AR targets, such as KLF Transcription Factor 9 (KLF9), KLF13, and PLZF have prominent roles in decidualization. For instance, loss of KLF9 expression in human endometrial stromal cells leads to compromised stromal function due to the premature expression of a key decidualizing factor, BMP2 [77]. Notably, one study has shown a reduction in endometrial BMP2 levels in infertile patients with PCOS [78]. BMP2 is induced downstream of progesterone/cAMP action in both mouse and human uterine stroma during decidualization. cAMP induces the expression of BMP2 in human endometrial stromal cells during in vitro decidualization [79]. Likewise, aberrant androgen signaling dysregulates decidualization via upregulated MAGEA11, impaired downstream targets, and interaction with the progesterone/cAMP pathways.

## 4. Inflammation and Immune Dysregulation in PCOS

PCOS is closely associated with persistent low-grade inflammation, marked by a skewed balance of pro-inflammatory and anti-inflammatory elements, including cytokines and immune cells [80]. This altered immune profile in PCOS influences not only metabolic functions but also reproductive capabilities. Inflammation is an indicator of early-stage endometrial disturbances in PCOS cases. Research examining the endometrial transcriptome during the implantation window in infertile obese women with PCOS has revealed significant disruptions in key inflammatory pathways. In particular, inflammation-related processes, especially those linked to TNFR1 signaling, were pronounced, indicating an innately inflamed state within the endometrial setting in PCOS [81]. Hyperandrogenism, another PCOS characteristic, can intensify this inflammation. High levels of androgens may stimulate the production of inflammatory cytokines, like TNF-α and interleukins, exacerbating the inflammatory environment of the endometrium [82].

### 4.1. Immune Cell Alterations in PCOS

In women with PCOS, hormonal imbalances and altered immune cell numbers within the endometrium create a challenging environment for implantation and pregnancy. Here, we discuss the roles and alterations of immune cells within the reproductive tracts of PCOS patients.

Natural killer (NK) cells are essential for immune defense and are barely detectable in the ovarian follicles of both PCOS patients and healthy individuals [83]. However, they are found to be significantly increased in peripheral blood in infertile women with PCOS [84]. These elevated peripheral NK cells are likely to be related to infertility or recurrent miscarriage [85]. Uterine NK (uNK) cells, a distinct subtype found within the endometrium, are characterized by their high expression of CD56 and low CD16 expression levels [86]. They play a critical role in pregnancy, particularly in the early stages of implantation and in the establishment of the placenta. However, in PCOS patients, there is a significantly lower percentage of CD56+/CD16− uNK cells in the late secretory endometrium, which may contribute to the condition’s reproductive complications [86].

Macrophages, derived from monocytes, are key immune cells found throughout human tissues that engage in various functions, including phagocytosis and both innate and adaptive immunity. Their activity ranges from pro-inflammatory to anti-inflammatory responses based on the surrounding tissue environment [87]. CD68+ and CD163+ macrophages are present throughout the menstrual cycle, peaking in the late luteal phase [88,89]. They are crucial for creating a receptive endometrium for embryo implantation and for the process of endometrial decidualization. An increase in CD68+ macrophages has been linked to miscarriage [90], and notably, in PCOS patients, there is an elevation of both CD68+ and CD163+ M2 macrophages. This increase may correlate with insulin resistance and a rise in inflammatory factors associated with PCOS [91].

Dendritic cells (DCs) are pivotal for successful pregnancy due to their role in antigen presentation, aiding the immune system in deciding whether to reject or tolerate a blastocyst [92]. Unique phenotypes of uterine DCs (uDCs) during the embryo-receptive phase contribute to this process, with specific subpopulations like DC-SIGN+ CD14+ CD83– DCs activating regulatory T cells. Key modulators such as heme oxygenase-1 and human chorionic gonadotropin, along with GM-CSF, regulate uDC maturation, essential for stress response, iron recycling, and immune regulation [92]. Deletion of DCs impairs uterine receptivity, tissue remodeling, and angiogenesis, leading to implantation failure and highlighting DCs’ critical role in decidualization [93]. The dynamic interaction between DCs and the uterine environment, influenced by cytokines like Interleukin-10 (IL-10) and hormones like estrogen and progesterone, modulates their function. This intricate balance of DC subsets, influenced by hormonal signals and the uterine microenvironment, is crucial for maintaining pregnancy, signifying the complex interplay between the immune system and endocrine factors in reproductive success [92]. The dysfunction of DCs, observed in normal-weight PCOS patients, may be linked to the syndrome’s pathogenesis. Granulocyte-macrophage colony-stimulating factor (GM-CSF) down-regulation in endometrial stromal fibroblasts of women with PCOS affects endometrial receptivity and DC cell migration [94].

While T cells are a major component of the lymphocytes in the non-pregnant endometrium, CD3+ T cells constitute a smaller subset, making up about 1–2% of all lymphoblast cells [92]. However, a higher prevalence of CD3+ T cells has been noted in the endometrium of women who have suffered miscarriages [90]. Similarly, women with PCOS show an increased proportion of CD3+ T lymphocytes in the endometrium during the late secretory phase, which could have implications for fertility and pregnancy outcomes [86].

### 4.2. Inflammatory Environment in PCOS

In the critical “window of implantation”, a balanced interplay between pro-inflammatory and anti-inflammatory forces is essential for developing a receptive endometrium [20]. Any imbalance can negatively influence both implantation and the outcome of the pregnancy. In PCOS, a complex inflammatory environment is underscored by increased levels of cytokines, such as tumor necrosis factor α (TNF-α) and IL-6, which are integral to the condition’s progression and its reproductive challenges [20]. These cytokines are key genes that activate and regulate the pro-inflammatory cascade crucial for interactions at the fetal-maternal interface. exacerbates this situation by stimulating macrophages to produce TNF-α, thereby raising TNF-α levels within the endometrium [95,96].

Moreover, cytokines and chemokines like IL-1 and vascular endothelial growth factor (VEGF) play pivotal roles in creating a receptive endometrium for pregnancy. Their reduced expression in PCOS suggests a decreased receptivity, which is further hampered by the dysregulation of the Toll-like receptor (TLR)-mediated NF-κB signaling pathway and an overexpression of cytokines such as IL-6, IL-8, IL-18, and CRP [94,97,98]. Such persistent inflammation may prime the endometrium in a way that adversely affects receptivity and the success of implantation [99]. The interplay between hormonal imbalances and metabolic disorders in PCOS synergistically exacerbates this pro-inflammatory state, further impeding endometrial function [91].

Overall, the complex interactions among cytokines, chemokines, hormonal fluctuations, and metabolic factors in PCOS culminate in an enhanced pro-inflammatory state that may hinder the endometrium’s receptivity, consequently impacting implantation success and pregnancy viability.

## 5. PCOS Models Induced by Direct Hormonal Interventions

Animal PCOS models are employed to provide insights into the underlying mechanisms and potential treatments for PCOS. Hyperandrogenism stands as a prominent clinical feature of PCOS in patients. As such, direct androgen treatment methods using agents like dehydroepiandrosterone (DHEA), dihydrotestosterone (DHT), or testosterone propionate (TP) are commonly utilized to reproduce PCOS-like phenotypes in animals. Next, we delve into rodent models that have been instrumental in studying the pregnancy events in PCOS.

### 5.1. DHEA

DHEA is an androgen primarily originating from the adrenal glands. It is observed to be elevated in women with PCOS [100]. Therefore, it has been widely used to induce PCOS in rodent models [101]. Given the potential for prenatal DHEA treatments to stimulate embryonic resorption [102], DHEA is applied to prepubertal rodents instead. Briefly, 25-day-old mice received daily injections of DHEA at a dosage of 6 mg/100 g for 20 days [18]. Different mouse strains, including C57BL/6 mice, CD1 mice, and BALB/c mice, have been employed in establishing the DHEA-induced PCOS model.

The C57/B6 mouse strain is one of the most frequently used models to study PCOS. Their genetic background and robust response to hormonal manipulations make them a choice model. The DHEA-induced C57/B6 mouse model displays notable ovarian dysfunction characterized by a significant increase in the number of cystic follicles and the thickness of the theca cell layer. Conversely, there is a marked decline in the number of corpora lutea and dominant follicles, accompanied by irregular estrous cycling [103]. Additionally, uterine examinations revealed considerable congestion and edema in DHEA-treated mice, with the uterine canal filled with hydrocele, suggesting an inflammatory state. Histological analysis further confirmed abnormal endometrial thickening and an expanded uterine luminal diameter. Furthermore, there was a pronounced elevation in the expression of F4/80 in both ovarian and uterine tissues, indicating heightened inflammation [103]. However, the C57/B6 strain presents a notable challenge; it displays strong uterine edema [103]. This physiological response makes it difficult to evaluate pregnancy-related events, encompassing uterine receptivity, decidualization, and placental development in PCOS. Consequently, this model emerges as most apt for probing into ovarian dysfunction and immune imbalances in PCOS.

In the DHEA-induced CD1 mouse model, PCOS ovaries were characterized by the presence of multiple cystic follicles, featuring a sizable fluid-filled antrum and degenerating granulosa cell layers [18]. Notably, while the number of blastocysts retrieved from PCOS mice was substantially reduced, their morphological appearance remained comparable to those from vehicle control mice. Embryo transfer was applied to exclude the effects from ovulation and embryos. When healthy embryos from healthy mice were transferred into PCOS pseudopregnant recipients, a marked decline in the implantation rate was evident, suggesting that the endometrial receptivity was compromised in PCOS mice [18]. A deeper investigation revealed that elevated estrogen levels, along with potential disruptions in the LIF-STAT3 pathway, might be the culprits behind the observed embryo implantation failures in PCOS mice. Moreover, there was a pronounced impairment in artificial decidualization in these mice [18]. Significantly, CD1 mice, devoid of uterine edema and possessing the capability for effective embryo transfer, offer a promising model for further exploration of pregnancy dynamics in PCOS.

The DHEA-induced BALB/c mouse model displays impaired oocyte quality [104]. This model exhibited both pro-inflammatory and pro-oxidant responses in the uterus, evident by elevated prostaglandin F2 alpha production and increased uterine nitric oxide synthase (NOS) activity [104]. Concurrently, there was a decline in antioxidant defenses, marked by reduced superoxide dismutase (SOD) and catalase (CAT) activities, and reduced glutathione (GSH) levels. Furthermore, there was a rise in CD4+ and a decline in CD8+ T lymphocytes in the uterine tissue [105,106]. These alterations might correlate with low implantation rates seen in polycystic ovary syndrome in women. However, the endometrial receptivity and the capability of stroma cells to undergo decidualization in this model remain unclear.

### 5.2. DHT

DHT, a potent non-aromatizable androgen, is converted from testosterone by 5α-reductase [107]. Elevated levels of DHT have been noted in PCOS women [108]. The DHT-induced PCOS rodent model successfully mimicked many clinical features of human PCOS, including ovarian cysts, hormonal imbalance, and insulin resistance, providing a robust platform for therapeutic investigations and mechanistic studies [101]. In this model, 20-day-old female rats or mice were implanted subcutaneously with a 90-day continuous DHT release pellet [109]. The DHT-induced C57/B6 mouse model exhibited reproductive anomalies, including anovulation and infertility [110]. In DHT-induced PCOS mice with a mixed background (C57/B6, CD1, 129Sv), despite a general trend toward infertility, a few were able to become pregnant. The offspring from these pregnancies experienced delayed puberty and displayed higher testosterone levels as they reached adulthood [101]. This model presents an opportunity to delve into the reproductive aspects of PCOS and the potential for epigenetic transmission of the syndrome. Recently, the PCOS-like pregnant rat model has facilitated a greater understanding of the complexities of pregnancy under the influence of PCOS. Briefly, Sprague-Dawley rats, upon exhibiting a vaginal plug indicative of GD 0.5, were administered daily doses of DHT (1.66 mg/kg) and insulin (6.0 IU) from GD 7.5 to 13.5 to induce a state of PCOS-like hyperandrogenism and insulin resistance [111]. Overall, this model aids in understanding the distinct challenges and physiological adjustments that occur during pregnancy in conditions similar to PCOS.

### 5.3. TP

Neonatal administration of TP in mouse models has shown promising potential in replicating certain PCOS-related ovarian phenotypes, particularly anovulation and polyfollicular ovaries [101]. Specific findings, such as the absence of corpora lutea and altered responses to hCG, suggest potential disruptions in follicular function and possible premature luteinization of follicles. These observed changes in mice resemble some of the follicular abnormalities seen in women with PCOS [112,113]. However, the suitability of this model for studying pregnancy events is questionable. One significant limitation arises from the neonatal administration of TP, which may induce developmental changes that do not accurately reflect the adult onset of PCOS in humans. This timing issue can lead to an altered hormonal milieu from a very early stage, potentially skewing the reproductive system’s development and function in ways that are not representative of PCOS-related fertility issues in adult women. Moreover, while the model effectively demonstrates certain aspects of the syndrome, such as the endocrine disturbances, it falls short in replicating the complex interplay of factors that affect pregnancy in PCOS women.

In conclusion, hormonally-induced rodent PCOS models are well-established experimental systems that have significantly contributed to our current understanding of PCOS. These models have several advantages, including the development of PCOS-like symptoms without the need for genetic manipulation or other endocrine disturbances. Nonetheless, it is important to note that these models do not fully replicate the human phenotype, particularly regarding the heterogeneity and chronic nature of the syndrome. Moreover, these rodent models primarily address the androgenic aspect of PCOS and may not encompass all the endocrine and metabolic nuances of the disorder. Despite these limitations, it has provided critical insights into the disease mechanism and has been pivotal in studying the pregnancy processes in PCOS.

## 6. Conclusions

PCOS presents significant challenges to pregnancy, impacting various stages from conception to gestation. This condition gives rise to hormonal imbalances and insulin resistance, and creates a compromised endometrial environment. PCOS patients exhibit impaired endometrial receptivity and compromised cyclic decidualization, stemming from dysregulated hormonal and molecular signaling pathways. Additionally, the dysregulation of the immune system and heightened inflammation introduce further complexity to the hormonal and metabolic disturbances in PCOS. These elements collectively contribute to a heightened susceptibility to pregnancy failure. The use of animal models that mimic the hormonal conditions of PCOS is highly valuable in enhancing our understanding of these issues. Furthermore, additional research regarding the relationships among the hormonal, metabolic, and immune systems in PCOS is crucial to develop more efficacious treatment strategies. Overall, advancements in PCOS research and treatment hinge on the combined efforts of medical professionals, scientists, and patients.

## Figures and Tables

**Figure 1 ijms-25-01203-f001:**
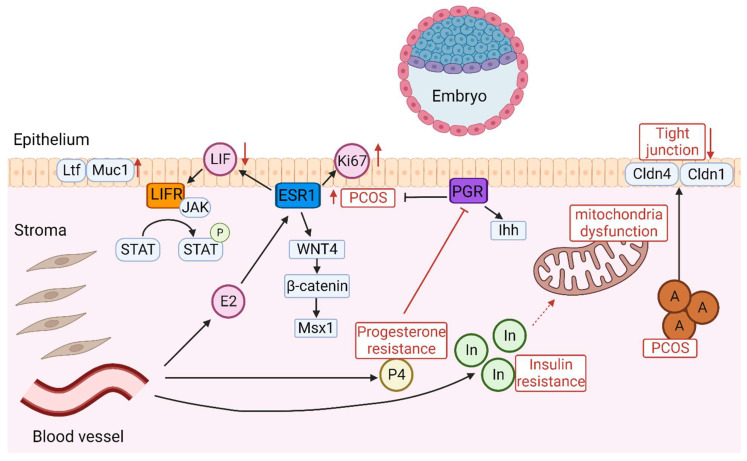
Diagram of the intricate signaling pathways involved in embryo implantation. In PCOS, abnormal expression of ESR1 increases Ki-67 expression in the endometrial epithelium and stroma, leading to decreased endometrial receptivity. LIF triggers the activation of JAK which in turn activates signal transducer and activator of STAT3 through phosphorylation. The role of the LIF-STAT3 pathway is characterized by inactivation in PCOS, which leads to the production of Ltf and Muc1, suggesting that the endometrium is in a non-receptive state. The dysregulation of the Wnt/β-catenin/Msx1 pathway impairs endometrial receptivity in PCOS and Msx1 is increased. Insulin resistance contributes to progesterone resistance and exacerbates issues with endometrial functionality. The abnormal expression of Ihh signaling in PCOS is depicted as a key factor affecting endometrial receptivity. Elevated androgen levels impact the integrity of tight junctions in the endometrial epithelium by decreasing Cldn4 and Cldn1. Insulin resistance impairs endometrial receptivity in PCOS, leading to mitochondrial dysfunction. A: Androgen; P4: Progesterone; E2: Estrogen; In: Insulin; P: phosphorylation.

**Figure 2 ijms-25-01203-f002:**
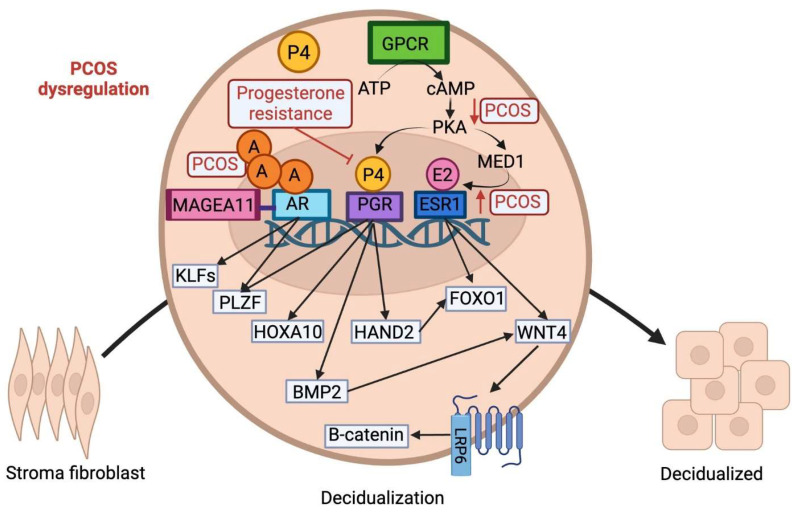
Schematic illustration of the molecular signaling pathways involved in the transformation of stromal fibroblast cells to epithelioid-like decidual cells. The decidual reaction is initiated via a complex interplay of progesterone, estrogen, cAMP, and androgen. Androgen hormone, P4, and E2 cross the cell membrane and translocate into the nucleus to bind to their respective associated receptors. Within the nucleus, the expression of both PGR and ESR1 is mediated by cAMP signaling via a G-protein coupled receptor. PGR regulates downstream expression of HAND2, FOXO1, HOXA10, and BMP2. PLZF, another component of the signaling cascade governing cell motility, is activated by both AR and PGR. The AR receptor is regulated via the co-regulator MAGEA11 and controls expression of KLF9, KFL12, and PLZF. Estrogen signaling via ESR1 controls downstream expression of WNT4 via MED1, a subunit of the mediator coactivator complex. WNT4 plays a crucial role in decidualization through binding to a seven-pass transmembrane Frizzled receptor and activating the B-catenin pathway. A: androgen; P4: progesterone; E2: estrogen.

## Data Availability

Not applicable.

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
