# Peer review of "Implantation and Decidualization in PCOS: Unraveling the Complexities of Pregnancy"

_ijms, 2024, doi:10.3390/ijms25021203_

Round 1

Reviewer 1 Report

Comments and Suggestions for Authors

The current review aims to describe how hormonal/metabolic imbalance affect endometrial receptivity and decidualization in PCOS patients. While the topic is of interest, this manuscript falls short of its promises. The major weakness of the review is that the authors did not provide over all conclusion and future aspects of the review.

1. Please describe some mechanistic viewpoint how hormonal/metabolic imbalance affect endometrial receptivity in PCOS in the abstract section also.

2.  Overall conclusion and future aspects of the review is missing. Please describe in the last section.

3.  Figure 1. LIF showed increased in PCOS. But authors described in line 84-85 LIF signaling decreased. Please describe why?

4. Line 380-384 says several studies have shown, but end of the line authors only cited one reference. Please correct accordingly.

5. Please describe current treatment strategy of PCOS in paragraph.

Author Response

Dear Reviewer 1,

We are very grateful for your positive response and enthusiasm for our manuscript entitled “Implantation and Decidualization in PCOS: Unraveling the Complexities of Pregnancy " by Matsuyama et al. (manuscript # ijms-2771268). We made a sincere effort to address your comments in the resubmission of our manuscript; we made substantial modifications to the text and revised Figure 1. We found your suggestions to be extremely helpful and, by making these requested changes, we feel that our manuscript is much stronger. Reviewer comments are in italics, and our responses are listed below in normal font.

Review 1

The current review aims to describe how hormonal/metabolic imbalance affect endometrial receptivity and decidualization in PCOS patients. While the topic is of interest, this manuscript falls short of its promises. The major weakness of the review is that the authors did not provide over all conclusion and future aspects of the review.

Thanks for your comments. We acknowledge the importance of providing an overall conclusion and discussing future aspects of the review. We have made the necessary revisions to improve the manuscript to meet your expectations.

  1. Please describe some mechanistic viewpoint how hormonal/metabolic imbalance affect endometrial receptivity in PCOS in the abstract section also.

Thanks for your suggestion. To enhance the comprehensiveness of the abstract, we have incorporated a mechanistic viewpoint explaining how hormonal and metabolic imbalances influence endometrial receptivity in PCOS.

  1.  Overall conclusion and future aspects of the review is missing. Please describe in the last section.

Thanks for your suggestion, we have included an overall conclusion and discussed future aspects of the review in the final section.

  1.  Figure 1. LIF showed increased in PCOS. But authors described in line 84-85 LIF signaling decreased. Please describe why?

We apologize for the mistake, and we have revised it. The levels of LIF were found to be reduced in PCOS. We have accordingly revised Figure 1 to accurately reflect this information.

  1. Line 380-384 says several studies have shown, but end of the line authors only cited one reference. Please correct accordingly.

Thank you for your attention to detail. We have added the reference to this sentence and corrected the text.

  1. Please describe current treatment strategy of PCOS in paragraph.

Currently, there is no cure for PCOS, with therapies centred on the patient's primary complaint. Treatment focuses on reducing hyperandrogenism symptoms, restoring menstrual regularity, and achieving conception. While our primary emphasis remains on the mechanistic aspects, it is important to briefly describe the ongoing treatment strategies for PCOS, which we have done in the introduction.

Best,

Shuyun Li

Reviewer 2 Report

Comments and Suggestions for Authors

The text delves into Polycystic Ovary Syndrome (PCOS), a widespread hormonal disorder affecting 5-15% of women globally. It explores the intricate aspects of PCOS and how it impacts pregnancy, revealing challenges in fertility due to hormonal imbalances and insulin resistance. Despite efforts to restore ovulation through medication, women with PCOS encounter lower pregnancy rates and heightened risks of implantation failure and miscarriage. The review goes deeper into the complexities of implantation in PCOS, shedding light on hormonal and metabolic imbalances that affect the uterus's readiness for pregnancy, including issues with estrogen signaling and the crucial process of endometrial decidualization. It also examines the roles of various immune cells and inflammatory processes in the uterus, contributing to the reproductive challenges associated with PCOS. Furthermore, the text discusses the use of rodent models to gain insights into PCOS, especially those induced by hormonal interventions. These models provide valuable information about how PCOS impacts pregnancy and potential avenues for treatment. In essence, the comprehensive review underscores the necessity for an advanced understanding of PCOS and the development of innovative treatment strategies to address the intricate interplay of hormonal, metabolic, and immune factors that contribute to reproductive complications in women with PCOS. Some weak points were listed point by point

Title: The title is well chosen, reflecting the study being reported.

Overall: The aim is well emphasized and explained. The paper is well written and was very pleasant to read. However, there is a lack of a broader view, synthesis and discussion of the facts presented. The discussion/conclusion section needs to be completed.

Abstract: It is well written, reflecting the study being reported.

Introduction :

The introduction section is attractive to read, emphasizing the reason for conducting this study.

V36: please describe more precisely the term “accelerated release”, since now it is unclear

V41: “ovarian cysts” is not good term form PCOS – there are follicles, and we talk about cyst when the size is >1-2cm.

 Implantation in PCOS

This section is well written, explaining correctly the issues.

Decidualization in PCOS

It is well written and clear. 

Considerations of progesterone action, progesterone concentrations and progesterone resistance should be supplemented with information on current knowledge on luteal phase deficiency diagnosis and treatment PMID: 33827766. The complexity of proper timing of progesterone administration - an important difficulty during therapy of PCOS. It is worth noting the correct timing of the inclusion of progesterone supplementation for example PMID: 36833150

Inflammation and Immune Dysregulation in PCOS and PCOS Models Induced by Direct Hormonal Interventions  - these sections are well written and no comments are necessary

"Conclusions" section for entire text, pointing main points is missing it should be completed

Author Response

Dear Reviewer 2,

We are very grateful for your positive response and enthusiasm for our manuscript entitled “Implantation and Decidualization in PCOS: Unraveling the Complexities of Pregnancy " by Matsuyama et al. (manuscript # ijms-2771268). We made a sincere effort to address your comments in the resubmission of our manuscript; we made substantial modifications to the text and revised Figure 1. We found your suggestions to be extremely helpful and, by making these requested changes, we feel that our manuscript is much stronger. Reviewer comments are in italics, and our responses are listed below in normal blue font.

Review 2

The text delves into Polycystic Ovary Syndrome (PCOS), a widespread hormonal disorder affecting 5-15% of women globally. It explores the intricate aspects of PCOS and how it impacts pregnancy, revealing challenges in fertility due to hormonal imbalances and insulin resistance. Despite efforts to restore ovulation through medication, women with PCOS encounter lower pregnancy rates and heightened risks of implantation failure and miscarriage. The review goes deeper into the complexities of implantation in PCOS, shedding light on hormonal and metabolic imbalances that affect the uterus's readiness for pregnancy, including issues with estrogen signaling and the crucial process of endometrial decidualization. It also examines the roles of various immune cells and inflammatory processes in the uterus, contributing to the reproductive challenges associated with PCOS. Furthermore, the text discusses the use of rodent models to gain insights into PCOS, especially those induced by hormonal interventions. These models provide valuable information about how PCOS impacts pregnancy and potential avenues for treatment. In essence, the comprehensive review underscores the necessity for an advanced understanding of PCOS and the development of innovative treatment strategies to address the intricate interplay of hormonal, metabolic, and immune factors that contribute to reproductive complications in women with PCOS. Some weak points were listed point by point

Thank you for providing a detailed summary of the review's main points and highlighting areas that need improvement. Your feedback is valuable in helping to refine and enhance the quality of the text.

Title: The title is well chosen, reflecting the study being reported.

Overall: The aim is well emphasized and explained. The paper is well written and was very pleasant to read. However, there is a lack of a broader view, synthesis and discussion of the facts presented. The discussion/conclusion section needs to be completed.

We appreciate your feedback and recognize the importance of providing an overall conclusion in the final section. We have included an overall conclusion and discussed future aspects of the review in the final section.

Abstract: It is well written, reflecting the study being reported.

 We appreciate your positive feedback on the abstract.

Introduction :

The introduction section is attractive to read, emphasizing the reason for conducting this study.

V36: please describe more precisely the term “accelerated release”, since now it is unclear

Thanks for your valuable feedback. The term "accelerated release" is not appropriated and we have changed it.

V41: “ovarian cysts” is not good term form PCOS – there are follicles, and we talk about cyst when the size is >1-2cm.

Thank you for pointing out the terminology issue. We revised the term "ovarian cysts" to accurately reflect the presence of follicles, especially emphasizing that the term "cyst" is typically used for structures larger than 1-2cm in size. This adjustment will ensure the accuracy of the terminology in describing the features of PCOS.

 Implantation in PCOS

This section is well written, explaining correctly the issues.

Decidualization in PCOS

It is well written and clear. 

We appreciate your acknowledgment that these sections are well-written.

Considerations of progesterone action, progesterone concentrations and progesterone resistance should be supplemented with information on current knowledge on luteal phase deficiency diagnosis and treatment PMID: 33827766. The complexity of proper timing of progesterone administration - an important difficulty during therapy of PCOS. It is worth noting the correct timing of the inclusion of progesterone supplementation for example PMID: 36833150

Thank you for providing additional insights and references related to progesterone action, concentrations, and resistance in PCOS. We have incorporated information on luteal phase deficiency diagnosis and treatment, as well as the importance of correct timing for progesterone supplementation in PCOS therapy, with reference to the provided PMIDs.

Inflammation and Immune Dysregulation in PCOS and PCOS Models Induced by Direct Hormonal Interventions  - these sections are well written and no comments are necessary

Thank you for your positive feedback.

"Conclusions" section for entire text, pointing main points is missing it should be completed

Thank you for highlighting the need for a "Conclusions" section. We have included an overall conclusion and discussed future aspects of the review in the final section.

Best,

Shuyun Li